# Efficient Production of Segmented Carbon Nanofibers via Catalytic Decomposition of Trichloroethylene over Ni-W Catalyst

**DOI:** 10.3390/ma16020845

**Published:** 2023-01-15

**Authors:** Arina R. Potylitsyna, Yuliya V. Rudneva, Yury I. Bauman, Pavel E. Plyusnin, Vladimir O. Stoyanovskii, Evgeny Y. Gerasimov, Aleksey A. Vedyagin, Yury V. Shubin, Ilya V. Mishakov

**Affiliations:** 1Boreskov Institute of Catalysis, Pr. Ac. Lavrentieva, 5, Novosibirsk 630090, Russia; 2Faculty of Natural Sciences, Novosibirsk State University, Str. Pirogova 2, Novosibirsk 630090, Russia; 3Nikolaev Institute of Inorganic Chemistry, Ac. Lavrentieva 3, Novosibirsk 630090, Russia

**Keywords:** nickel, tungsten, carbon erosion, trichloroethylene, carbon nanofibers, carbon nanomaterials

## Abstract

The catalytic utilization of chlorine-organic wastes remains of extreme importance from an ecological point of view. Depending on the molecular structure of the chlorine-substituted hydrocarbon (presence of unsaturated bonds, intermolecular chlorine-to-hydrogen ratio), the features of its catalytic decomposition can be significantly different. Often, 1,2-dichloroethane is used as a model substrate. In the present work, the catalytic decomposition of trichloroethylene (C_2_HCl_3_) over microdispersed 100Ni and 96Ni-4W with the formation of carbon nanofibers (CNF) was studied. Catalysts were obtained by a co-precipitation of complex salts followed by reductive thermolysis. The disintegration of the initial bulk alloy driven by its interaction with the reaction mixture C_2_HCl_3_/H_2_/Ar entails the formation of submicron active particles. It has been established that the optimal activity of the pristine Ni catalyst and the 96Ni-4W alloy is provided in temperature ranges of 500–650 °C and 475–725 °C, respectively. The maximum yield of CNF for 2 h of reaction was 63 g/g_cat_ for 100Ni and 112 g/g_cat_ for 96Ni-4W catalyst. Longevity tests showed that nickel undergoes fast deactivation (after 3 h), whereas the 96Ni-4W catalyst remains active for 7 h of interaction. The effects of the catalyst’s composition and the reaction temperature upon the structural and morphological characteristics of synthesized carbon nanofibers were investigated by X-ray diffraction analysis, Raman spectroscopy, and electron microscopies. The initial stages of the carbon erosion process were precisely examined by transmission electron microscopy coupled with elemental mapping. The segmented structure of CNF was found to be prevailing in a range of 500–650 °C. The textural parameters of carbon product (S_BET_ and V_pore_) were shown to reach maximum values (374 m^2^/g and 0.71 cm^3^/g, respectively) at the reaction temperature of 550 °C.

## 1. Introduction

The creation of carbon nanomaterials (CNM) with different structures, ranging from graphene and fullerene to nanofibers, seems to be a promising direction in the field of materials science. In terms of practice, the greatest interest is focused on an improvement of synthetic procedures to produce carbon nanotubes (CNT), carbon nanofibers (CNF), as well as diverse CNM-based composites with desired characteristics. Filamentous carbon materials (CNT and CNF) can be used as reinforcing additives in cement stone and concrete [1], antifriction agents for lubricants and oils [2], as fillers in polymer matrices to impart electrical conductivity and to increase the strength and resistance to abrasion [3], and also as carriers for catalysts used in the hydrodechlorination [4] and selective hydrogenation [5] processes.

The principle industrially relevant method for the synthesis of CNF is based on catalytic pyrolysis of hydrocarbons using metal catalysts (known as CCVD—Catalytic Chemical Vapor Deposition) [6,7]. As a rule, catalytic systems for the CCVD process, as well as for many other reactions (e.g., hydrogen evolution reaction), have to contain dispersed active particles, preparation of which usually requires passing through a number of stages [8]. There are a number of ways to disperse the initial catalytic systems in order to increase their specific surface area and improve the catalytic activity [9,10,11]. One possible option to simplify the preparative procedure is to apply the so-called “self-dispersion” method directly in the reactor, during the contact of metallic precursor with the reaction mixture. This approach is based on the process of carbon erosion (CE) or metal dusting (MD), which is well-known as a negative phenomenon leading to the destruction of iron- and nickel-based alloys operating in industrial reactors at temperatures of 400–800 °C in a carbon-containing atmosphere. Spontaneous disintegration of a bulk or coarse-dispersed metal (alloy) is accompanied by the formation of numerous submicron particles, which catalyze the growth of graphite-like carbon filaments [12,13,14]. For example, the complete wastage of 1 g of nichrome (Ni-Cr) in contact with 1,2-dichloroethane vapors produces about 10^14^ active particles with an average diameter of 250 nm, on which the CNF growth takes place [15]. Thereby, the target use of the carbon erosion phenomenon makes it possible to combine the stage of obtaining the catalyst in an active form and the synthesis of CNF, which greatly facilitates the overall procedure.

CO, CH_4_, and C_2+_ hydrocarbons and their mixtures as well as the chlorinated hydrocarbons can serve as a carbon-containing source in the CCVD process [16,17,18]. Catalytic processing of organochlorine compounds is of particular research interest because it might solve a difficult environmental problem related to toxic waste disposal [19,20]. As known from the literature, the supported Ni-containing catalysts are the most commonly used for the pyrolysis of such chlorinated hydrocarbons as 1,2-dichloroethane (1,2-DCE) and trichloroethylene (TCE). At the same time, there are only a few works devoted to the use of TCE as a carbon source for CNF synthesis [21]. Our recent study related to the catalytic pyrolysis of TCE has demonstrated that the maximum yield of the resulting carbon product is 2.6 times greater than that obtained by the decomposition of 1,2-DCE under identical conditions [22]. Thus, the search for an effective catalyst and the study of features of the catalytic pyrolysis of TCE is appeared to be an important scientific task.

It should be emphasized that the catalytic processing of chlorinated hydrocarbons over the self-dispersing Ni-catalysts makes it possible to obtain carbon nanomaterial with unique structural peculiarities [23,24]. An opportunity to produce N-doped CNF materials via combined pyrolysis of the chlorinated hydrocarbons and N-containing precursors has been recently demonstrated as well [25,26]. The implementation of the periodic “chlorination-dechlorination” process on the active metallic surface was found to be responsible for the formation of a segmented secondary structure of growing carbon filaments [27]. The resulting carbon product is characterized by a high specific surface area (up to 400 m^2^/g) and pore volume (up to 1 cm^3^/g), which makes this material promising for various adsorption and catalytic applications [28,29,30].

Among the most frequently used metals to catalyze the synthesis of CNM (Fe, Co, and Ni) [31,32,33], nickel and its alloys demonstrate the greatest activity and resistance to deactivation in the case of catalytic pyrolysis of organochlorine substrates [34]. Meanwhile, the addition of the modifying metal M (up to 10 wt%) to the Ni-M catalyst can have a noticeable promoting impact on the catalytic performance of nickel. For example, Ni-Mo (up to 8 wt% Mo) and Ni-W (4 wt% W) alloys exhibited the greatest activity in the decomposition of 1,2-DCE: the yield of CNF increased by 1.5–2.5 times if compared to pure nickel [35,36]. The observed positive effect is associated primarily with an increase in the rate of decomposition of the chlorinated hydrocarbons. On the other hand, the replacement of nickel by larger Mo and W atoms results in the formation of the solid solutions Ni-M with an increased crystal lattice, which ultimately leads to an enhancement of the carbon capacity of the Ni-M alloy [37,38,39]. It should be noted that the opportunity of using the Ni-W alloy system as a catalyst for hydrocarbon pyrolysis remains practically unrevealed. At the same time, very recent works report a promising effect of tungsten on the activity of nickel catalysts in the synthesis of carbon nanofibers [40] and nanotubes [41]. Therefore, of particular interest is the exploration of the impact of the small addition of W (4 wt%) upon the catalytic performance of a self-dispersing Ni-catalyst. The choice of the promoter metal concentration was based on the results of previous research, in which the optimal catalyst composition for the pyrolysis of 1,2-DCE to produce CNF has been defined [36]. Moreover, the possible high activity of Ni-W catalysts in CCVD of chlorinated hydrocarbons makes this approach very attractive from the ecologic point of view (processing of the organochlorine wastes).

The aim of the present work was to estimate the efficacy of catalytic decomposition of TCE (C_2_HCl_3_), containing the unsaturated C=C bond and characterized by the intermolecular chlorine-to-hydrogen ratio of 3:1, over metallic Ni and Ni-W catalysts. TCE was selected as an insufficiently studied, accessible substrate known as a principal constituent in a number of organochlorine wastes. The synthesized carbon nanomaterials were thoroughly characterized by a set of physicochemical methods, including X-ray diffraction analysis, Raman spectroscopy, and transmission electron microscopy coupled with elemental mapping. Special attention was paid to the early stages of the carbon erosion process when catalytically active particles are being formed. The unique segmental structure of the carbon product was characterized by scanning transmission electron microscopies. The textural characteristics were measured by the low-temperature nitrogen adsorption technique. In addition, the results of comparative longevity tests for 96Ni-4W and 100Ni catalysts are also presented.

## 2. Materials and Methods

### 2.1. Materials and Reagents

The following chemicals used for the synthesis of catalysts were purchased from Vekton (Saint-Petersburg, Russia): H_2_WO_4_ (pure), ammonia solution (25%, high purity grade), and acetone (chemically pure). The precursor salt [Ni(NH_3_)_6_]Cl_2_ was synthesized as described elsewhere [42]. Chemically pure TCE (Komponent-reactive, Moscow, Russia), high-purity argon, and hydrogen were used in the catalytic experiments. All the gases were of chemical purity grade and were used without any preliminary purification.

### 2.2. Synthesis of 100Ni and 96Ni-4W Catalysts

A calculated amount of H_2_WO_4_ (0.054 g) was added to 10 mL of 25% ammonia solution, heated (60–70 °C) with stirring until almost complete evaporation of the solution, and left for a day. Next, 20 mL of 25% ammonia solution was added to H_2_WO_4_ and heated with stirring until the powder dissolved. The solution was pale yellow and cloudy. 10 mL of H_2_O and Ni(NH_3_)_6_Cl_2_ taken in a certain ratio (3.789 g) were added to the resulting solution, stirred until complete dissolution, and cooled to room temperature. The solution was poured into 300 mL of acetone and cooled down to T~0 °C with stirring. The resulting sediment of light violet color was filtered, washed abundantly with acetone, and dried at room temperature for 10 h. The dried sample was then reduced in a hydrogen flow of 130 mL/min at 800 °C for 1 h. The reduced 96Ni-4W alloy sample was cooled down to room temperature in a helium flow. A pure nickel catalyst (100Ni, reference sample) was prepared by a similar procedure, excluding the addition of H_2_WO_4_.

### 2.3. Studies on the Metal Dusting Process and Carbon Deposition

The catalytic studies were performed in a flow-through quartz reactor equipped with McBain balances, thus allowing one to follow the accumulation of the carbon product over the catalyst in a real-time mode [43]. The weight of the initial bulk alloy (2.0 ± 0.05 mg) was placed in a quartz basket, which was hooked to a calibrated quartz spring. Before the experiment, the reactor was purged with argon (150 mL/min) and heated to the reaction temperature (450–725 °C) in an argon flow. After that, the sample was reduced in a hydrogen flow (100 mL/min) until the catalyst weight was stabilized. Next, the reaction mixture TCE/Ar/H_2_ (TCE—6 vol%, Ar—56 vol%, H_2_—38 vol%) was fed to the reactor. The total flow rate of this mixture was 267 mL/min. Each experiment lasted for 2 h. In the case of the longevity tests, the experiments continued for 7 h. During the experiment, the cathetometer was used to follow the extension of the quartz spring caused by the process of carbon deposition and then to calculate the weight gain with a time on stream. At the end of the experiment, the reactor was cooled in an argon flow to room temperature. The carbon product was unloaded and weighed in order to measure the carbon yield (Y_C_, g/g_cat_).

### 2.4. Characterization of Catalysts and Carbon Nanomaterials

The powder X-ray diffraction (XRD) analysis of 96Ni-4W and pure Ni samples has been performed at room temperature on a Shimadzu XRD-7000 diffractometer (Shimadzu, Tokyo, Japan) using CuKα radiation, and graphite monochromator. The patterns were recorded in the step mode within the angular range 2θ = 15–80°, step 0.1° (survey diffraction pattern) and 2θ = 140–148°, step 0.05° (for precise determination of a lattice parameter). Data from the PDF database were used as references [44]. The lattice parameters were determined by the position of 331 diffraction reflection (at 2θ ≈ 144°) using the PowderCell 2.4 software (BAM, Berlin, Germany) [45]. The volume-averaged crystallite sizes were calculated from the broadening of the (111), (200), and (220) peaks using the Scherrer equation [46], after the separation of the contribution from the instrumental broadening. The deconvolution and fitting of the X-ray diffraction lines based on the Pearson (PVII) function were performed using the WinFit 1.2.1 software (Institute of Geology and Mineralogy, Erlangen, Germany) [47].

The chemical composition of synthesized 96Ni–4W alloy was determined by inductively coupled plasma atomic emission spectroscopy (ICP-AES) on a Thermo Scientific iCAP-6500 spectrometer (Thermo Fisher Scientific Inc., Waltham, MA, USA). Prior to the measurement procedure, the sample was dissolved in a mixture of nitric and hydrofluoric acid.

Raman spectra of obtained carbon nanomaterials were recorded on a Horiba Jobin Yvon LabRAM HR Ultraviolet-Visible-Near Infrared (UV-VIS-NIR) Evolution Raman spectrometer (Horiba, Kyoto, Japan) equipped with Olympus BX41 microscope (Olympus Corp., Tokyo, Japan) and 514.5-nm line of Ar ion laser. In order to avoid the thermal decomposition of the sample, the power of light focused on a spot with a diameter of ~2 μm was less than 0.8 mW.

The secondary structure and morphology of pristine catalysts and synthesized carbon nanomaterials were examined by scanning electron microscopy (SEM) on a JSM-6460 instrument (JEOL Ltd., Tokyo, Japan) at magnifications of 1000× to 100,000×.

Additionally, the morphology of the carbon nanomaterials was examined on a two-beam scanning electron microscope TESCAN SOLARIS FE-SEM (TESCAN, Brno, Czech Republic) working with an acceleration voltage of 20 kV in a secondary electron mode.

The transmission electron microscopy (TEM) studies coupled with elemental mapping were performed using a Hitachi HT7700 TEM (acceleration voltage 100 kV, W source, Hitachi Ltd., Tokyo, Japan) equipped with a STEM system and a Bruker Nano XFlash 6T/60 energy dispersive X-ray (EDX) spectrometer (Bruker Nano GmbH, Berlin, Germany).

The textural characteristics of the obtained carbon nanomaterials were determined by low-temperature nitrogen adsorption, Brunauer–Emmett–Teller (BET) method. The adsorption/desorption isotherms were measured at 77 K on an automated ASAP-2400 (Micromeritics, Norcross, GA, USA) device. The temperature of preliminary degassing of carbon nanomaterial samples was 300 °C.

## 3. Results and Discussion

### 3.1. Study of Microdispersed 100Ni and 96Ni-4W Catalysts by XRD and SEM

The samples of the synthesized catalysts were explored by SEM and XRD methods. As can be seen from SEM images (Figure 1), both samples have a porous structure, which is built from the fused particles of ~1–2 μm in diameter connected by bridges. The specific surface area of the pristine samples is rather developed and achieves the value of 10 m^2^/g. It was found that the addition of W has no effect on the morphology and structure of the nickel catalyst. According to ICP-AES analysis data, the W content in the composition of 96Ni-4W alloy was about 4.4 wt%.

The phase composition of the synthesized 96Ni-4W and pure Ni samples was examined by a powder XRD analysis. The XRD patterns recorded in various 2θ ranges are shown in Figure 2. In the diffraction patterns within 2θ = 15–100°, a set of reflections typical of a face-centered cubic (fcc) lattice can be observed (Figure 2a). Note that no impurity peaks were identified. The reflections for the 96Ni-4W sample are seen to be shifted to the low-angle range with respect to pure nickel (100Ni). The observed shift of the peaks is more pronounced in the far-angle range (2θ = 140–150°). Thus, the developed synthetic procedure allows obtaining the single-phase alloy having the lattice parameter (a) of 3.529 Å (for comparison, a = 3.524 Å for 100Ni). The observed data, along with the absence of any extra peaks in the XRD patterns, permits one to claim that the prepared bimetallic 96Ni-4W sample is represented by a single-phase solid solution. It is worth noting that the addition of W leads to a certain broadening of the (331) reflection (Figure 2b), which is associated with a decrease in the crystallite size. The crystallite sizes for the 96Ni-4W and 100Ni samples calculated from the XRD data were found to be 35 and 70 nm, respectively.

### 3.2. Catalytic Decomposition of TCE over 100Ni and 96Ni-4W Catalysts

The synthesized 100Ni and 96Ni-4W microdispersed samples were studied in the catalytic pyrolysis of TCE vapors in an excess of hydrogen to produce CNM. The reaction temperature was varied within a range from 475 to 725 °C. The results were plotted as kinetic curves representing the dependence of the carbon nanomaterial weight gain on the time of exposure to the reaction mixture (Figure 3). At the end of the experiment, the yield of CNM (or catalyst productivity) was calculated as a ratio of the weight of the synthesized carbon product to the weight of the catalyst sample used (g/g_cat_).

The choice of the temperature interval used for the catalytic pyrolysis of TCE is explained by the following reasons. Both the upper and lower temperature limits are determined by a rapid deactivation of the catalyst. At temperatures below 500 °C, the decomposition of TCE is suppressed due to irreversible chlorination of the Ni surface (to form NiCl_2_), while at T > 700 °C the catalyst deactivation is caused by an encapsulation of the alloy surface with amorphous carbon deposits. In both cases, the yield of carbon product does not exceed the level of 10 g/g_cat_. The observed regularities are in good agreement with the reported results of previous works devoted to the study of temperature regimes of the carbon erosion of bulk metals and alloys [35,48].

It can be clearly seen from Figure 3 that the rate of accumulation of the carbon product is insignificant at the initial stage (first 10–20 min). A typical “delay”, known in the literature as the “induction period” (IP), is explained by a slow process of carbon erosion of the alloy [49]. In this regard, all the kinetic curves of the carbon product accumulation can be nominally divided into 2 stages: (i) induction period, and (ii) sustainable growth of CNM. The value of the carbon product yield of 5 g/g_cat_ (which corresponds to a 500% increase in the weight of the catalyst loading) was conventionally chosen as the boundary separating these two stages. As will be shown further, the time to reach a 500% weight gain (i.e., the duration of IP) depends on the catalyst composition and the reaction temperature.

During the IP of the reaction, the initial alloy undergoes disintegration under the action of carbon erosion, which is accompanied by the emergence of numerous active dispersed particles. Simultaneously, the nucleation of the graphite-like phase and the subsequent growth of carbon filaments on top of the active particles are observed in course of the first phase. The intense growth of the carbon nanomaterial (the beginning of a significant increase in the sample weight) is considered the transfer to the second stage. In order to characterize the second stage of the interaction, such parameters as the carbon product yield (*P*, g/g_cat_) and the carbon deposition rate (*v*, mg/min) can be used. The *v* parameter was calculated as the slope of the kinetic curve in a region of 20–120 min. The values of these parameters calculated for 100Ni and 96Ni-4W samples at different temperatures are summarized in Table 1. It is worth noting that the experiments for the reference sample (100Ni) were carried out at four temperature points (500, 550, 600, and 650 °C), which made it possible to compare 100Ni with 96Ni-4W alloy and establish the effect of the tungsten addition on the catalytic performance of nickel. As for the 96Ni-4W alloy catalyst, its behavior in the TCE decomposition was thoroughly studied at some extra temperature points.

Figure 4 shows the dependence of IP duration on temperature for both catalyst samples. As can be seen from the plot, for the 96Ni-4W alloy, the duration of IP varies insignificantly (10–15 min) in a temperature interval of 550–700 °C. This indicates that the carbon erosion process proceeds at approximately the same rates within a wide temperature range. In turn, for the monometallic reference sample (100Ni), the minimum duration of IP at 600 °C was equal to 20 min, which is 2 times higher than that for the 96Ni-4W alloy. At higher temperatures (650 °C), it takes a much longer time (40 min) for a pure nickel sample to disintegrate, which testifies to a much lower rate of the CE process. The observed effect can be explained by the deposition of amorphous carbon due to the decomposition of TCE on the outer surface of metal agglomerates, which leads to a surface blockage and a decrease in the rate of carbon transfer into the bulk of nickel. In the case of the 100Ni monometallic sample, the high-temperature deactivation occurs at T~650 °C. Meanwhile, the introduction of W into the Ni-alloy results in a noticeable rise in the deactivation temperature (up to 700 °C). It should be noted that the IP duration for 96Ni-4W alloy has very close values in the temperature range of 550–700 °C. Most probably, the addition of W makes the alloy more resistant to chlorination at lower temperatures, and this fact determines an increase in the CE rate.

The process of carbon erosion of the 96Ni-4W sample was precisely studied by TEM coupled with elemental mapping (Figure 5 and Figure 6). Since the CE process proceeds very rapidly in the TCE atmosphere, the temperature of the reactor was decreased to acquire the possibility to register all the changes taking place at the initial stages. This allows deceleration of the decomposition of the substrate and the disintegration of the catalyst. A temperature of 500 °C was chosen for this purpose. At this temperature, the IP duration exceeds 30 min (Figure 4), and no catalyst deactivation by amorphous carbon is observed. Therefore, in these experiments, the 96Ni-4W was exposed to the reaction mixture for 2, 5, 10, and 15 min. The 2-h experiment represents the steady-state operation of the catalyst.

As is seen from Figure 5a, after 2 min of exposure, the surface of the microdispersed alloy particles, which serve as a precursor of the active catalyst, undergoes loosening and disintegration under the action of the carbon nanostructures being formed. Some small particles of up to 100 nm in diameter are detached from the bulk of the sample. At this stage, the carbon product has a fibrous structure, and the diameter of the fibers varies in a range from 10 to 200 nm. Figure 5d–f shows the TEM images for the sample after 5 min of exposure. The catalyst particles of spherical or oval shape not exceeding 10 nm in size are evidently seen. No agglomerated areas of the initial bulk alloy are seen. Thereby, a complete disintegration of the precursor with the formation of a variety of active particles can be stated. After 10 min of the CE process, the larger particles (100–250 nm in size) catalyzing the growth of the nanostructured carbon material are observable. Supposedly, they were formed via the sintering of smaller particles that appeared after the first 5 min of CE. As a rule, one of the faces of such large particles is responsible for the process of TCE decomposition. Next, carbon diffuses to the other face where the release and graphitization of carbon take place. The relatively small particles of the active component (up to 50 nm in size) are also seen. Some of them are separated from the large ones and incorporated into the structure of the growing fibers (Figure 5i). Therefore, it can be concluded that during the first 10 min of exposure, active catalytic particles of optimal size are being formed. Such particles provide an efficient growth of CNF. After 15 min, the majority of thus formed particles operate in a steady-state regime (Figure 5j–l).

According to the elemental mapping of the samples, nickel and tungsten are evenly distributed over the samples. The chlorine species appeared on the surface of the samples exposed to the reaction mixture for 15 min and more. Based on EDX analysis data it is possible to conclude that the concentration of chlorine in the obtained CNF samples does not exceed the value of 1.2 at%. The observed fact is in good agreement with earlier reported data obtained by EDX and XPS methods for CNF materials produced via catalytic pyrolysis of DCE over Ni-M alloys [26,35,48]. It can also be seen from Figure 6c that chlorine species are mainly adsorbed on the surface of active metal particles, where they appear to bind preferentially to tungsten atoms. The observed fact might have a significant impact on the enhanced catalytic performance of the Ni-W system with respect to decomposition of chlorinated hydrocarbons.

In the next step, a comparative analysis of the second stage of the TCE decomposition over the studied catalysts was performed. Figure 7 shows a diagram demonstrating the dependence of the 2-h carbon product yield on the temperature. As can be seen from the experimental results, the maximum productivity of pure 100Ni was 63 g/g_cat_ at a temperature of 600 °C. It should be reminded that the minimum duration of IP (20 min) for pure nickel is also observed at this temperature point. Meanwhile, the maximum yield of CNF for the 96Ni-4W catalyst was as high as 112 g/g_cat_, which is almost two times higher than that for the unmodified nickel. As can be seen from the data presented in Figure 7, the alloyed Ni-W catalyst outperforms the reference sample (pure nickel) in productivity within the entire 475–725 °C temperature range. Based on the literature data concerning the decomposition of 1,2-dichloroethane (C_2_H_4_Cl_2_) at the same conditions, one can find that the maximum yield, in this case, did not exceed 45 g/g_cat_ [35]. It is also worth emphasizing that the productivity of the 96Ni-4W alloy at temperatures from 575 to 650 °C remains very high and changes very slightly (102 ± 10 g/g_cat_). The observed fact might be taken into account when scaling the process up since the lowest possible temperature is one of the important criteria to ensure the efficient realization of the TCE processing.

Thereby, a comparison of the catalytic activity of 100Ni and alloyed 96Ni-4W catalyst revealed that even a small amount of tungsten (4 wt%) has a significant positive impact upon the catalyst performance of nickel, which is reflected by a remarkable shortening of the IP duration as well as by an increase in the carbon product growth rate (Table 1).

As it was mentioned above, there is a lack of information in the scientific literature concerning the impact of tungsten on the catalytic performance of Ni-based catalysts used for the decomposition of hydrocarbons and their derivatives [40,41]. In the recently reported paper [41] it was claimed that W is capable of increasing the catalytic activity and stability of nickel particles during the hydrocarbon decomposition due to the partial transfer of electrons from Ni to W, along with the formation of the W_2_C phase. The latter serves as a regulator of carbon atoms from the surrounding atmosphere to Ni. These assumptions can also be applied to the systems studied in this article. Meanwhile, it seems reasonable to consider W as an analog to Mo. In turn, Mo is very well known in the literature as one of the most effective promotors of Ni-catalysts used for the production of carbon nanotubes and nanofibers [22,50]. The emergence of a strong synergistic effect can be explained by the ability of Mo to enhance greatly the carbon capacity of nickel and to accelerate the diffusion of carbon during the CNM growth. The revealed impact of W addition is very significant in the case of TCE decomposition, thus showing a good potential for the Ni-W system to be further applied for the processing of polychlorinated aliphatic hydrocarbons.

### 3.3. Study of Morphology and Structure of Carbon Product

#### 3.3.1. XRD Data

As mentioned above, the solid-phase product of the catalytic pyrolysis of C_2_HCl_3_ vapors is represented by a nanostructured carbon material grown on dispersed active particles originating from the disintegration of the pristine samples. The results of the XRD analysis of the carbon materials are presented in Figure 8. In the diffraction patterns of studied samples, the graphite-like phase can be identified as the predominant phase (2Θ~26°). The second phase is represented by the dispersed metal particles (Ni or Ni-W alloy) present in the sample after the reaction. In addition, there are traces of the Ni_3_C phase registered for the CNF samples, obtained at 500–550 °C (Figure 8). The presence of nickel carbide is consistent with the supposed mechanism of the catalytic growth of carbon nanofibers known as the “carbide cycle mechanism” [51].

The content of metal particles of the catalyst within the composition of the samples was estimated as a ratio of the peak (111) area for the *fcc* phase at 2θ of 44.9° to the peak (002) area for graphite-like phase at 2θ of 25.9° (S_111_/S_002_). The biggest amount of the residue catalyst was found in the sample obtained via the decomposition of TCE over the 96Ni-4W catalyst at 650 °C (Table 2). It is worth noting that the lattice parameter for the samples obtained over pure nickel (100Ni) does not expectedly change at varying temperature conditions of the reaction. For the samples synthesized using 96Ni-4W alloy, the lattice parameters are different due to a big measurement error. In the cases of CNF@96Ni-4W(500) and CNF@96Ni-4W(550) samples, no reflection (331) in a 2θ range of 140–148°, which is used for the most precise estimation of the lattice parameter, was identified. Therefore, another reflection (220) in a 2θ range of 75–78° was used for the calculations. Moreover, the CNF@96Ni-4W(500) sample contains a very low amount of the metallic phase (S_111_/S_002_ = 0.03). This complicates the calculation of the lattice parameter using the reflection (220) as well and contributes to the overall uncertainty of measurement.

#### 3.3.2. Raman Data

Raman spectra of the carbon product formed over the 96Ni-4W(T) samples, where T is the synthesis temperature, for a region of the bands of first and second orders are presented in Figure 9. The first-order spectra are characterized by the G bands at ~1590–1600 cm^−1^, corresponding to allowed vibrations E_2g_ of the hexagonal lattice of graphite [52], and by the disorder-induced D line of an activated A_1g_ mode due to finite crystal size [53,54] at ~1340 cm^−1^.

The D_2_ bands at ~1618 cm^−1^, corresponding to the disordered graphitic lattice (surface graphene layers, E_2g_ symmetry) [55], appeared for the samples obtained at temperatures of 600 °C and above. The bands D_3_ at ~1500 cm^−1^ and D_4_ at ~1200 cm^−1^, assigned to the amorphous carbon and the disordered graphitic lattice (A_1g_ symmetry) or polyenes [56] and typical for soot and related carbon materials, are present in spectra of all the samples.

Among the group of second-order bands, the most intensive ones are the bands 2D at ~2677 cm^−1^ and D + D_2_ at ~2927 cm^−1^. The other bands, 2D_2_ and G*~D_4_ + D, are of noticeably lower intensity. In order to approximate the second-order lines, excluding the 96Ni-4W(475) and 96Ni-4W(500) samples, a set of bands 2D and D + D_2_ of lower intensity and sufficiently higher half-width should be used. It is worth noting that for comparative analysis of the data obtained by Raman spectroscopy and other methods, it should be taken into account that the information provided by Raman spectroscopy for carbon materials corresponds to a laser penetration depth of ~0.1–0.2 μm [57].

The dependences of the main parameters (I_D_/I_G_, I_D3_/I_G_, and half-width HWHM G) on the temperature of the synthesis are demonstrated in Figure 10a. As is seen, an increase in the synthesis temperature results in a rise in the I_D_/I_G_ ratio. Considering the equation I_D_/I_G_ = C’(λ)·L_a_^2^ proposed by Ferrari and Robertson [54], where C’ is about 0.0055 for the wavelength of 514.5 nm, this corresponds to an increase in the in-plane crystallite sizes (L_a_) from 13.4 to 17.44 Å. The decrease in HWHM G, which usually occurs simultaneously with the increase in the in-plane crystallite sizes (L_a_), shows an ill-defined minimum at 600–650 °C followed by a rise at 700 °C. The dependence of the amorphous carbon portion (I_D3_/I_G_ ratio) on the synthesis temperature has an explicit minimum at 550–600 °C.

Since, in our case, the samples were obtained at close temperatures, the described diversity is not quantitative. This is mostly a qualitative difference defined by the catalytic process of CNF growth. Therefore, there are three temperature points, which should be marked out in terms of the catalytic growth of CNF over 96Ni-4W catalysts: 475–500 °C, near 600 °C, and near 700 °C.

Raman spectra of the carbon product formed over the 100Ni(T) samples for T = 500, 590, and 620 °C are shown in Figure 11. The corresponding dependences of the main parameters on the synthesis temperature are summarized in Figure 10b. As can be seen, these samples are characterized by a higher portion of the amorphous carbon (I_D3_/I_G_ ratio) at 500 °C if compared with the case of 96Ni-4W. An increase in temperature leads to the rapid growth of the in-plane crystallite sizes (L_a_) along with a decrease in the amorphous carbon portion (I_D3_/I_G_ ratio). However, this process is also accompanied by a deceleration of the catalytic reaction leading to its complete cessation at 650 °C. This allows concluding that the observable phenomenon of carbon ordering is connected with the decelerated carbon growth near the edge of the temperature window.

#### 3.3.3. SEM and TEM Data

The morphology and structure of the carbon product obtained at different reaction temperatures were examined by scanning and transmission electron microscopies (Figure 12, Figure 13 and Figure 14). The material resulting from the catalytic decomposition of TCE over 100Ni and 96Ni-4W is predominantly represented by long carbon nanofibers. Figure 12 compares SEM and TEM images of the carbon fibers obtained at the same temperatures over different catalytic systems. It can be seen that the morphology of CNM is almost independent of the catalyst composition. The secondary structure of carbon filaments is characterized by a segmental arrangement regardless of the catalyst’s composition.

High-resolution TEM images show in more detail the morphology of the carbon nanomaterials (Figure 13). It can be seen that both samples are represented by alternating graphene packets of different densities. The diameter of the fibers ranges from 100 to 250 nm.

According to TEM data presented in Figure 14, the catalyst is represented by rounded particles of submicron size (200–400 nm) connected to the grown carbon filaments. Depending on the geometric shape of the particle, the growth of CNF occurs simultaneously in 2–4 directions. It should be noted that the catalyst composition has a very slight effect on the morphology of the resulting carbon product.

At the same time, the main factor influencing the morphological and structural features of carbon fibers is the temperature of the TCE decomposition. For example, one can see that the process of carbon erosion was not accomplished at T = 475 °C: only the surface layer of the metal was subjected to fragmentation to form the active particles (Figure 14a). As mentioned above, this is due to the almost complete chlorination of metallic surface happening in course of the TCE decomposition at low temperatures. As the reaction temperature increases, the growth of carbon filaments accelerates but their structure becomes more defective. Thus, most of the fibers produced at T = 500–550 °C have rather regular segmented structures (Figure 14b,c), while the rise of temperature by 50 °C results in a growth of filaments with noticeably “damaged” segmentation (Figure 14d). This is accompanied by a growth of short fragments and the appearance of smaller catalyst particles (50 nm), which are most likely the product of the secondary fragmentation of large submicron particles. The metallic particles derived from the secondary disintegration also play the role of growth centers for the thinner carbon filaments.

A further increase in the reaction temperature (650 °C) leads to a higher extent of disordering and defectiveness of the resulting carbon product. In this case, the short lateral branches appear within the structure of carbon filaments, the contribution of non-segmental fibers increases, and the direction of the fiber growth becomes tortuous and winding, making it difficult to trace the beginning and the end of a single filament (Figure 14e,f). Finally, the view of CNM obtained at the maximum reaction temperature (700 °C) testifies to the presence of the partially amorphous product (Figure 14f).

According to the data of low-temperature N_2_ adsorption (BET method), the specific surface area and pore volume of the carbon material produced at 600 °C were 360 m^2^/g and 0.68 cm^3^/g (for the 100Ni sample) and 354 m^2^/g and 0.68 cm^3^/g (for the 96Ni-4W alloy). The closeness of values of the textural parameters allows one to infer that the composition of the catalyst used for TCE pyrolysis has no significant effect on the properties of the resulting carbon nanomaterial.

### 3.4. Results of Longevity Tests of 100Ni and 96Ni-4W Catalysts

In this study, the longevity (resource) tests of the catalysts were carried out at 600 °C for 7 h. The experiments were performed in a gravimetric flow-through setup equipped with McBain balances. The results of the test are shown in Figure 15 and Table 3. It can be seen that the carbon deposition rate for the 100Ni catalyst gradually decreases. It is only 11 mg/min by the end of the 7th hour, which is almost 4 times less than the rate at the initial stage. The alloyed 96Ni-4W catalyst also exhibited a certain decrease in the carbon deposition rate, but at the end of the test, it was equal to 39 mg/min, which is only 2.2 times less than the initial value (Table 3). In summary, the results of the longevity tests showed that, after the 7-h experiment, the rate of carbon accumulation over the 96Ni-4W alloyed catalyst is 3 times higher compared to the reference sample. The productivity of the 96Ni-4W alloy catalyst at the end of the test was as high as 256 g/g_cat_, which is 2 times superior with respect to the 100Ni reference sample (114 g/g_cat_).

## 4. Conclusions

In the present work, the catalytic decomposition of TCE over microdispersed 100Ni and 96Ni-4W catalysts was studied in a wide temperature range for the first time. Such a process results in the formation of carbon nanomaterial represented by long nanofibers attached to catalytic particles. The obtained carbon nanofibers possess a segmented secondary structure, regardless of the catalyst composition. Such a structure provides both a high specific surface area (360 m^2^/g) and a pore volume (0.68 cm^3^/g). It was revealed by Raman spectroscopy that the ordering degree of the carbon product strongly depends on the temperature of the process and the composition of the catalyst.

A precise study of the initial stages of the carbon erosion process (disintegration of the microdispersed particles under the action of the aggressive carburizing atmosphere) has shown that the formation of catalytically active particles of optimal size occurs during the first ten minutes. Next, the formed particles operate in a steady-state regime. Elemental mapping of the samples exposed to contact with the reaction medium for different times confirmed the uniform distribution of nickel and tungsten and detected the appearance of chlorine species on the surface of the samples after 15 min of interaction.

Additionally, a significant promoting effect of the tungsten addition was found, which is manifested in: (i) a 2-fold shortening of the IP duration; (ii) a 3-fold increase in the carbon deposition rate; and iii) an increase in the carbon productivity from 63 g/g_cat_ to 101 g/g_cat_ (600 °C, 2 h). The longevity tests revealed that the Ni-W alloy showed much better tolerance to deactivation, demonstrating the extremely high yield of 256 g/g_cat_ for 7 h, which is two times higher than that for the unmodified nickel.

## Figures and Tables

**Figure 1 materials-16-00845-f001:**
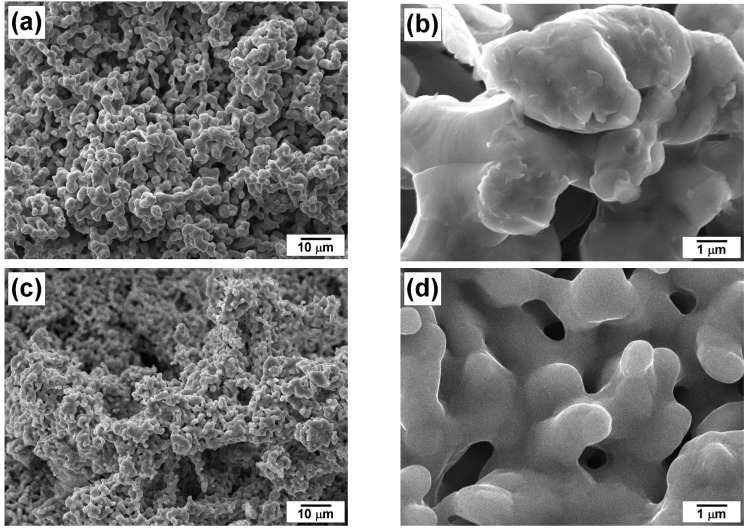
SEM images of the pristine samples prepared by co-precipitation and subsequent reductive thermolysis: (**a**,**b**) 100Ni; (**c**,**d**) 96Ni-4W.

**Figure 2 materials-16-00845-f002:**
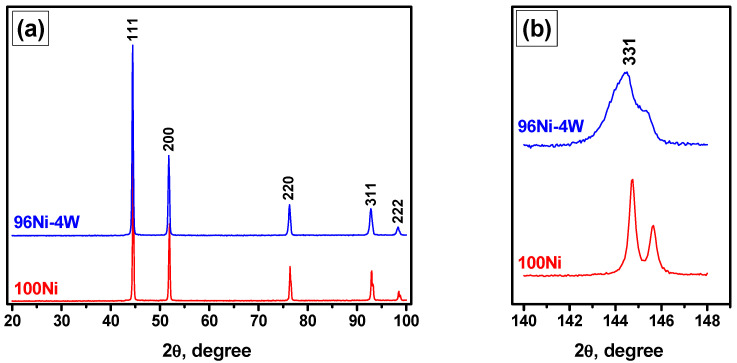
XRD profiles for 96Ni-4W alloy and 100Ni: survey patterns (**a**) and 331 reflections in the far angle range (**b**).

**Figure 3 materials-16-00845-f003:**
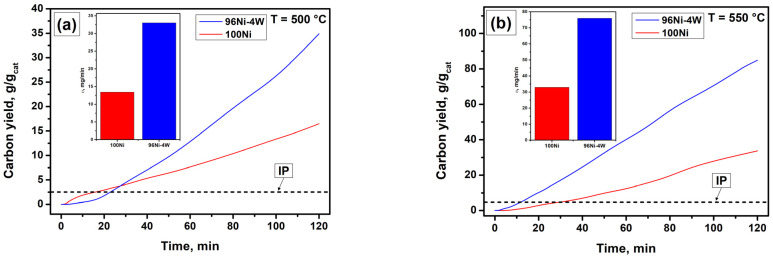
Carbon yield and corresponding carbon deposition rate (*v*, mg/min) during the 2-h decomposition of TCE vapors in excess of H_2_ over 100Ni and 96Ni-4W catalysts at temperatures: (**a**) 500 °C; (**b**) 550 °C; (**c**) 600 °C; (**d**) 650 °C.

**Figure 4 materials-16-00845-f004:**
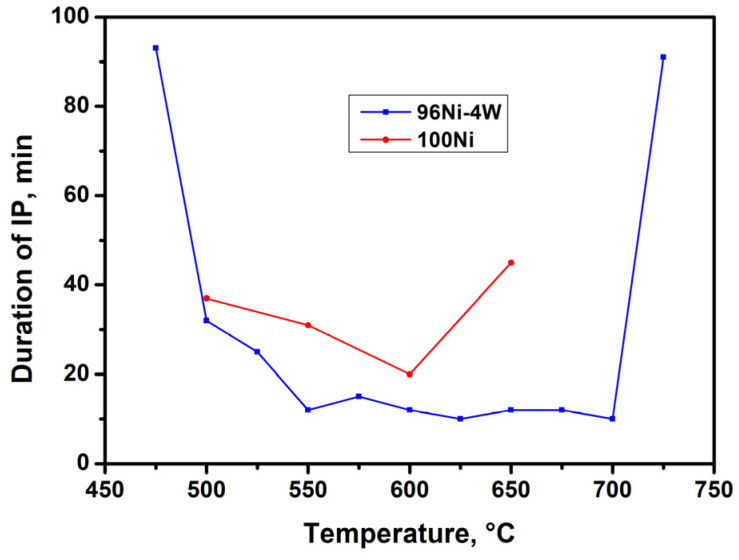
Dependence of IP duration on temperature for 100Ni and 96Ni-4W catalysts tested in the reaction of catalytic decomposition of TCE in excess of H_2_.

**Figure 5 materials-16-00845-f005:**
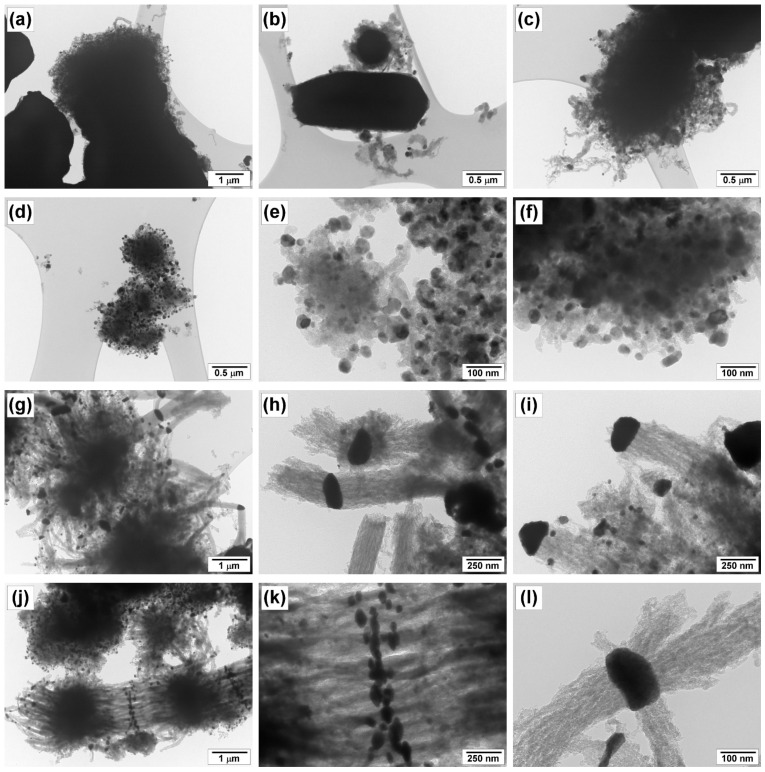
TEM images of the carbon nanomaterials and the catalytic particles formed as a result of carbon erosion of 96Ni-4W alloy during its interaction with TCE at 500 °C for 2 min (**a**–**c**), 5 min (**d**–**f**), 10 min (**g**–**i**), and 15 min (**j**–**l**).

**Figure 6 materials-16-00845-f006:**
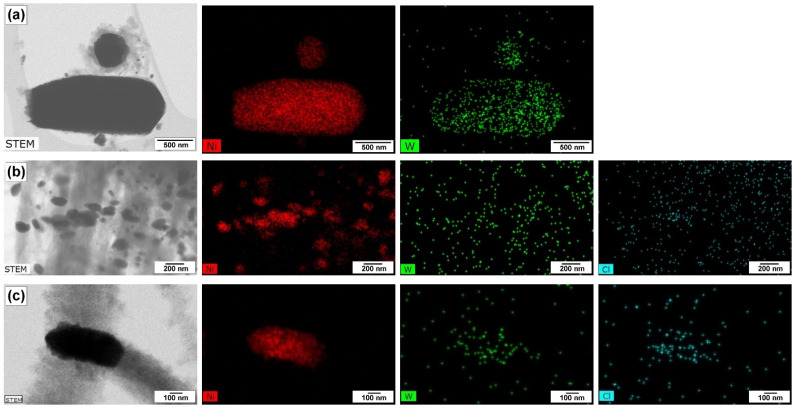
Elemental mapping of the catalytic particles formed as a result of carbon erosion of 96Ni-4W alloy during its interaction with TCE at 500 °C for 2 min (**a**), 15 min (**b**), and 2 h (**c**).

**Figure 7 materials-16-00845-f007:**
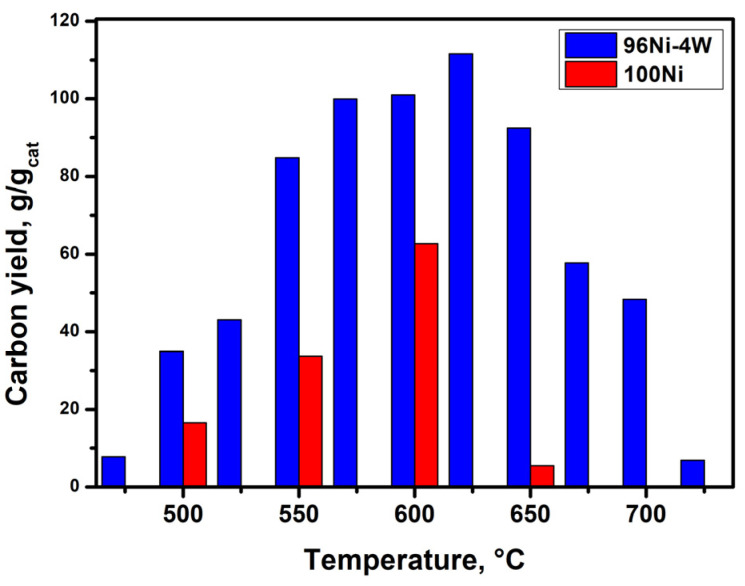
Dependence of the CNF yield on the temperature for 100Ni and 96Ni-4W catalysts tested in the reaction of catalytic decomposition of TCE in an excess of H_2_.

**Figure 8 materials-16-00845-f008:**
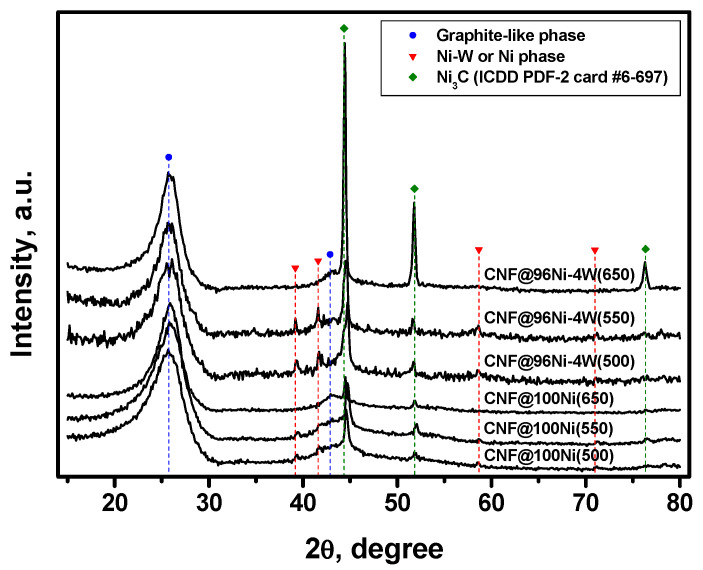
X-ray diffraction profiles for 96Ni–4W alloys and pure 100Ni samples after 2h reaction with TCE (500, 550, and 650 °C).

**Figure 9 materials-16-00845-f009:**
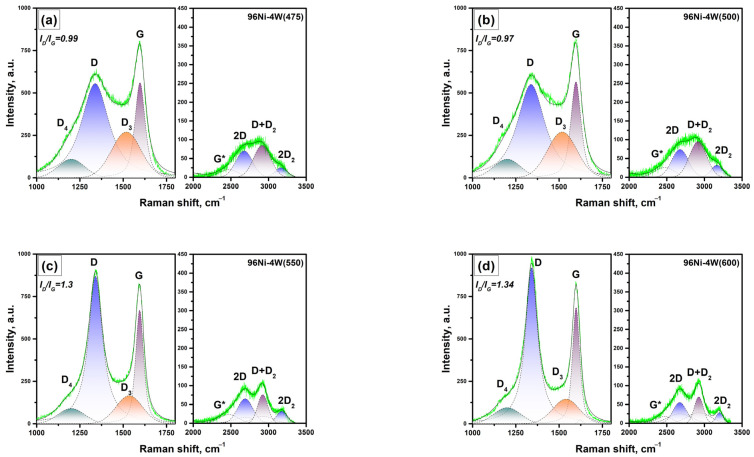
Raman spectra of the carbon product formed over the 96Ni-4W samples for a region of the bands of first and second orders: (**a**) 96Ni-4W(475); (**b**) 96Ni-4W(500); (**c**) 96Ni-4W(550); (**d**) 96Ni-4W(600); (**e**) 96Ni-4W(650); (**f**) 96Ni-4W(700).

**Figure 10 materials-16-00845-f010:**
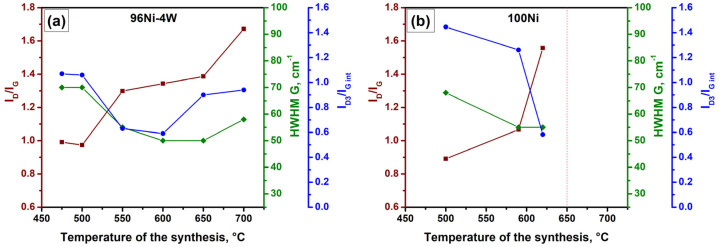
Dependences of the parameters (ID/IG, ID3/IGint, and HWHM G) on the synthesis temperature for the samples 96Ni-4W (**a**) and 100Ni (**b**).

**Figure 11 materials-16-00845-f011:**
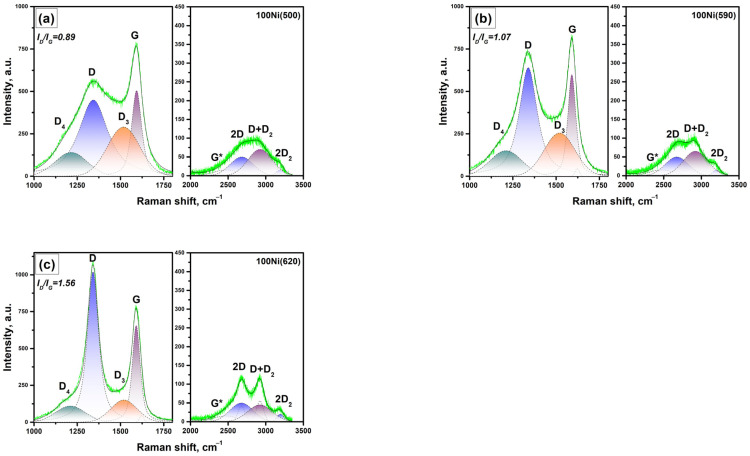
Raman spectra of the carbon product formed over the 100Ni samples for a region of the bands of first and second orders: (**a**) 100Ni(500); (**b**) 100Ni(590); (**c**) 100Ni(620).

**Figure 12 materials-16-00845-f012:**
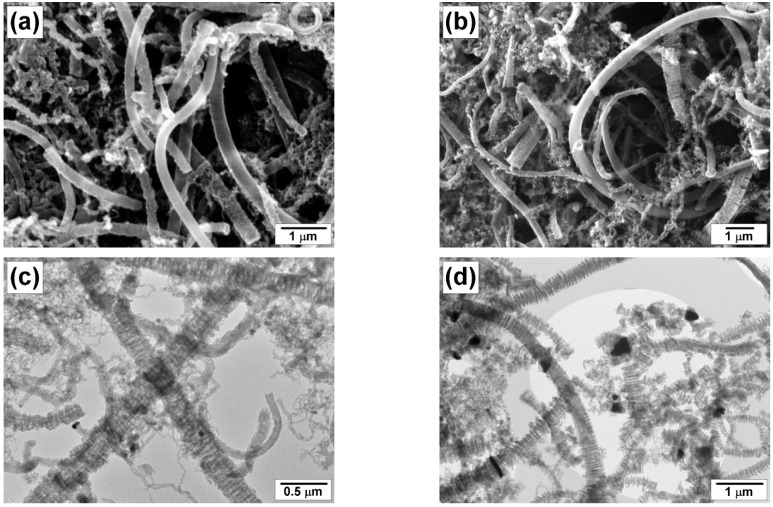
SEM (**a**,**b**) and TEM (**c**,**d**) images of the carbon product obtained by the decomposition of TCE at 600 °C over catalysts: (**a**,**c**) 100Ni; (**b**,**d**) 96Ni-4W.

**Figure 13 materials-16-00845-f013:**
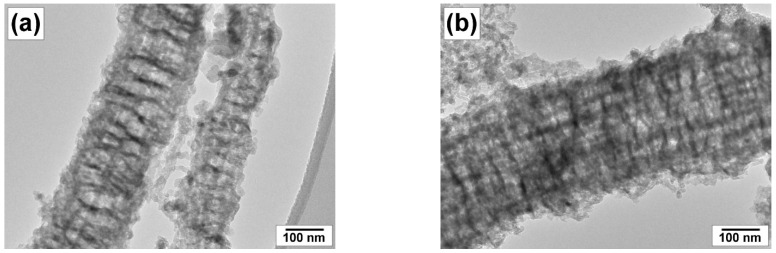
HR TEM images of the carbon product obtained by the decomposition of TCE at 600 °C over catalysts: (**a**,**b**) 100Ni; (**c**,**d**) 96Ni-4W.

**Figure 14 materials-16-00845-f014:**
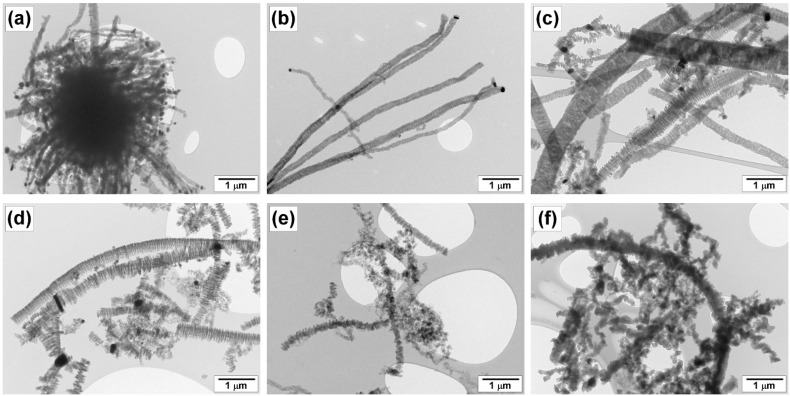
TEM images of the carbon product obtained via decomposition of TCE over 96Ni-4W alloy at temperatures: (**a**) 475 °C; (**b**) 500 °C; (**c**) 550 °C; (**d**) 600 °C; (**e**) 650 °C; (**f**) 700 °C.

**Figure 15 materials-16-00845-f015:**
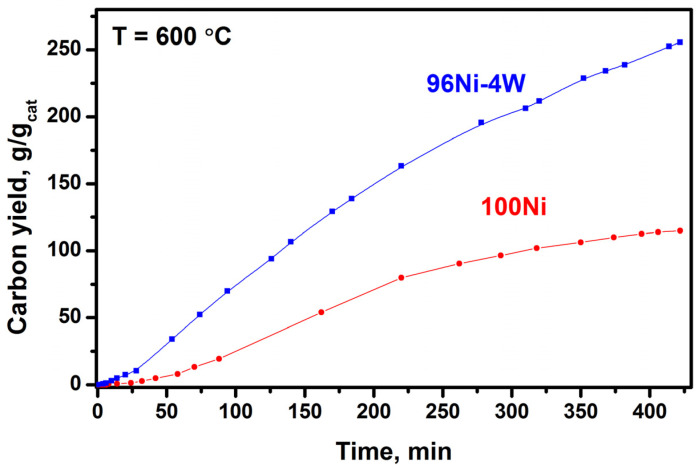
Accumulation of the carbon product during the decomposition of TCE over 100Ni and 96Ni-4W catalysts for 7 h at 600 °C.

**Table 1 materials-16-00845-t001:** Catalytic decomposition of TCE over 100Ni and 96Ni-4W samples within the temperature range of 450–650 °C.

N	Temperature, °C	100Ni			96Ni-4W		
		τ(IP), min	Yield of CNF, g/g_cat_	*v*, mg/min	τ(IP), min	Yield of CNF, g/g_cat_	*v*, mg/min
1	475	-	-	-	93	8	9
2	500	37	16	13	32	35	33
3	525	-	-	-	25	43	40
4	550	31	34	33	12	85	76
5	575	-	-	-	15	100	92
6	600	20	63	57	12	101	92
7	625	-	-	-	10	112	99
8	650	40 *	5	-	12	92	81
9	675	-	-	-	12	58	47
10	700	-	-	-	10	48	35
11	725	-	-	-	91	7	-

* deactivation.

**Table 2 materials-16-00845-t002:** XRD data for the carbon nanomaterials produced over 100Ni and 96Ni-4W samples in a temperature range of 500–650 °C.

Sample	Carbon				Metal Particles (Ni or Ni-W)			*S*_111_/*S*_002_
	*S*_002_ (*)	*d*_002_, Å	*D*, nm	n	*a*, Å	*D*, nm	*S* _111_	
CNF@100Ni(500)	11400	3.46	1.7	5	3.526(2) (140–148°) *3.522(3) (75–78°)*	25	600	0.05
CNF@100Ni(550)	9500	3.41	1.8	5	3.525(2) (140–148°) *3.523(3) (75–78°)*	25	500	0.05
CNF@100Ni(650)	6400	3.44	2.3	6	3.526(2) (140–148°) *3.527(3) (75–78°)*	>100	100	0.02
CNF@96Ni-4W(500)	2400	3.42	1.8	5	*3.541(4) (75–78°)*	40	70	0.03
CNF@96Ni-4W(550)	1300	3.45	2.7	7	*3.533(3) (75–78°)*	30	120	0.09
CNF@96Ni-4W(650)	5400	3.43	2.5	7	3.529(2) (140–148°) *3.531(3) (75–78°)*	35	1000	0.19

* S is the peak area.

**Table 3 materials-16-00845-t003:** Parameters of the TCE decomposition reaction carried out over 100Ni and 96Ni-4W catalysts for 7 h at 600 °C.

Reaction Time, min	100Ni		96Ni-4W	
	Yield of CNF, g/g_cat_	*v*, mg/min	Yield of CNF, g/g_cat_	*v*, mg/min
120	35	35	91	88
180	61	44	136	77
240	85	44	174	64
300	99	23	204	51
360	108	15	232	49
420	114	11	256	39

## Data Availability

Data are contained within the article.

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
