# Peer review of "Efficient Production of Segmented Carbon Nanofibers via Catalytic Decomposition of Trichloroethylene over Ni-W Catalyst"

_materials, 2023, doi:10.3390/ma16020845_

Round 1
Reviewer 1 Report
This contribution does not deserve publication at the present status.
The main drawback of the paper is the lack of the novelty and of the scientific content of the paper. In fact, this contribution could be considered as a short extension of some of previous papers of the authors, using very similar compounds (e.g. Mishakov et al.; Catalytic properties of bulk (1–x)Ni–xW alloys in the decomposition of 1,2-Dichlo-roethane with the production of carbon nanomaterials. Kinet. Catal. 2022, 63, 75-86).
The kinetic analysis of the
results presented on table 1 and Figure 6 is quite insufficient. Firstly, the
authors should indicate how they calculate the data of reaction rate from the
data on figures 3 and 9. The application of a rigorous kinetic model to analyse
these data, will allow to calculate the evolution of the reaction rate with the
time for the samples studied.
According to the data on Table 1, the Arrhenius plots allow an apparent activation energy of 47.6 kJ/mol for the 100Ni sample, and of 47.3 kJ/mol for the 96Ni-4W sample (see Figure A of the attached file). Therefore, there is not any difference between the values of the apparent activation energies and consequently there is not a promoting effect of W on the Ni-W catalyst.
Furthermore, an inspection of the data corresponding to the 96Ni-4W catalyst, indicates that at high temperatures (HT), above 550 ºC, there are strong diffusional limitations (see Figure B on the attached file). In this HT zone, the apparent activation energy is Ea,app=11.6 kJ/mol, a very low value, representative of severe external mass transfer limitations. On the contrary, at low temperatures, the Ea,app=75.5 kJ/mol, that is even higher than the observed value for the 100Ni sample. The authors should explain and discuss this fact in the manuscript.
In addition, it is necessary to complete the characterization of the catalytic materials after reaction (e.g. XRD, Raman, etc).
¿Why is the activity increased in presence of W? ¿Why does the carbon morphology not change in presence of W?
In summary, the authors should explain what is the true role of the W on the Ni-W catalyst.

Author Response
Q1. This contribution does not deserve publication at the present status. The main drawback of the paper is the lack of the novelty and of the scientific content of the paper. In fact, this contribution could be considered as a short extension of some of previous papers of the authors, using very similar compounds (e.g. Mishakov et al.; Catalytic properties of bulk (1–x)Ni–xW alloys in the decomposition of 1,2-Dichloroethane with the production of carbon nanomaterials. Kinet. Catal. 2022, 63, 75-86). A1. Thank you very much for your comment! We can understand your criticism very well. At the same time, it should be stressed that there is lack of information in scientific literature concerning the impact of tungsten on catalytic performance of Ni-based catalysts used for decomposition of hydrocarbons and their derivatives. We believe that the investigation of Ni-W alloys in catalytic pyrolysis would help to fill this gap. Moreover, the Ni-catalysts doped with W demonstrate really high activity in CCVD of chlorinated hydrocarbons, which makes them very attractive from the ecologic point of view (processing of the organochlorine wastes). Despite the seeming similarity of 1,2-dichloroethane (DCE) and trichloroethylene (TCE), there are several important differences: TCE molecule contains C=C bond and is characterized by significantly higher intramolecular ratio α = [H]/[Cl]. For DCE (C2H4Cl2) α = 4/2 = 2, whereas for TCE (C2HCl3) α = 1/3. Thus, for further elaboration of the effective catalyst, it seems to be important to study the peculiarities of TCE decomposition over the promising Ni-W catalyst, which was successfully tested for DCE processing. The revealed impact of W addition was even higher in case of TCE decomposition, thus showing a good potential for Ni-W system to be applied for processing of polychlorinated aliphatic hydrocarbons. According to your comment, the necessary emphases have been added to the text of manuscript. Q2. The kinetic analysis of the results presented on table 1 and Figure 6 is quite insufficient. Firstly, the authors should indicate how they calculate the data of reaction rate from the data on figures 3 and 9. The application of a rigorous kinetic model to analyse these data, will allow to calculate the evolution of the reaction rate with the time for the samples studied. According to the data on Table 1, the Arrhenius plots allow an apparent activation energy of 47.6 kJ/mol for the 100Ni sample, and of 47.3 kJ/mol for the 96Ni-4W sample (see Figure A below). Therefore, there is not any difference between the values of the apparent activation energies and consequently there is not a promoting effect of W on the Ni-W catalyst. Furthermore, an inspection of the data corresponding to the 96Ni-4W catalyst, indicates that at high temperatures (HT), above 550 ºC, there are strong diffusional limitations (see Figure B). In this HT zone, the apparent activation energy is Ea,app=11.6 kJ/mol, a very low value, representative of severe external mass transfer limitations. On the contrary, at low temperatures, the Ea,app=75.5 kJ/mol, that is even higher than the observed value for the 100Ni sample. The authors should explain and discuss this fact in the manuscript. A2. We are really grateful to your valuable comment and your efforts to replot the kinetic data! It made us reconsider completely this part of the manuscript. You are right, there is an evident mass transfer limitation for the Ni-W catalyst at high temperatures, which could be explained by its higher catalytic performance as compared with Ni, and it must be taken into account when plotting the graphs. At the same time, we are now convinced that there are insufficient kinetic data to make a rigorous comparison of Ea for Ni and Ni-W catalysts, since we don’t have enough temperature points made for Ni (the intermediate temperature points at 525 and 575°C are missed). Having not enough experimental data, it’s impossible to define the clear boundary between the kinetic and diffusion regime for both catalysts, which is crucial for the proper approximation. Thus, the more detailed and comprehensive study is needed to make such calculations possible. We plan to fulfil it in the future. That is why we have decided to eliminate Figure 6 (Calculation of the apparent activation energy) and all calculated Ea values from the text of paper. We believe that the exclusion of these data does not affect the conclusions in any way. Meanwhile, the positive effect of W addition remains undoubtful: it helps to enhance the rate of carbon deposition and boost the productivity of nickel. Thank you once again for your careful consideration! Q3. In addition, it is necessary to complete the characterization of the catalytic materials after reaction (e.g. XRD, Raman, etc). A3. Thank you for your suggestion! XRD data Following your comment, we have carried out the XRD analysis of the carbon materials obtained via catalytic pyrolysis of TCE over Ni and Ni-W catalysts (600°C, 2 h). The results are presented in Fig.1. In the diffraction patterns of studied samples, the graphite can be identified as the predominant phase (2Θ ~ 26°). The second phase is represented by the dispersed metal particles (Ni or Ni-W alloy) present in the sample after reaction. In addition, there are the traces of the Ni3C phase registered in both CNF samples (Fig.1). The presence of nickel carbide is consistent with the supposed mechanism of the catalytic growth of carbon nanofibers known as the “carbide cycle mechanism” [1]. Figure 1. X-ray diffraction profiles for 96Ni–4W alloy and pure 100Ni after 2h reaction with TCE (600 °C). The new Figure with XRD data has been added to the manuscript, along with the description. [1] Buyanov, R. A., Chesnokov, V. V., Afanas'ev, A. D., Babenko, V. S. Carbide mechanism of formation of carbonaceous deposits and their properties on iron-chromium dehydrogenation catalysts. Kinet. Catal. 1977, 18(4), 839-845. Raman spectroscopy data The Raman spectra for both CNF samples (Ni and Ni-W, 600°C, 2 h) demonstrate the similar features reported earlier for the segmented carbon nanofibers obtained on various Ni-M alloys [2, 3]. As found by comparing the obtained spectra, a slight increase in the amount of ordered carbon is observed in the case of W-containing catalyst (the ID/IG ratio changes from 0.9 (Ni) to 1.3 (Ni-W)). The proportion of amorphous carbon also decreases (ID3/G ~ 0.37 → 0.22). However, the cluster diameter change insignificantly (12.3 → 15.4 Å). Nevertheless, we consider this data to be redundant for this paper and did not insert it into the text. We believe that more detailed study of CNF materials produced at different temperatures by Raman spectroscopy should be accomplished in the next research. [2] Bauman, Y.I.; Rudneva, Y.V.; Mishakov, I.V.; Plyusnin, P.E.; Shubin, Y.V.; Korneev, D.V.; Stoyanovskii, V.O.; Vedyagin, A.A.; Buyanov, R.A. Effect of Mo on the catalytic activity of Ni-based self-organizing catalysts for processing of dichloroethane into segmented carbon nanomaterials. Heliyon 2019, 5, e02428. [3] Bauman, Y.I.; Shorstkaya, Y.V.; Mishakov, I.V.; Plyusnin, P.E.; Shubin, Y.V.; Korneev, D.V.; Stoyanovskii, V.O.; Vedyagin, A.A. Catalytic conversion of 1, 2-dichloroethane over Ni-Pd system into filamentous carbon material. Catalysis Today 2017, 293, 23-32. TEM data The TEM micrographs taken at higher magnifications have been added in Figure 6 of manuscript in order to highlight the primary structure of the produced segmented nanofibers. Once again, the comparison of the TEM data reveals almost no impact of W addition on structural peculiarities of carbon nanofibers. Figure 2. TEM images of the carbon product obtained by the decomposition of TCE at 600 °C over catalysts: (e) 100Ni; (f) 96Ni-4W. Q4. ¿Why is the activity increased in presence of W? ¿Why does the carbon morphology not change in presence of W? In summary, the authors should explain what is the true role of the W on the Ni-W catalyst. A4. This is very important question, thank you! As it was mentioned above, there is lack information in literature concerning the effect of W on catalytic performance of Ni (Co, Fe) in the decomposition of hydrocarbons to produce carbon nanomaterial. Meanwhile, it seems reasonable to consider W as the analogue to Mo. In turn, Mo is very well known in the literature as one of the most effective promotor to Ni-catalysts used for production of carbon nanotubes and nanofibers [4, 5]. The emergence of strong synergistic effect can be explained by the ability of Mo to enhance greatly the “carbon capacity” of nickel and to accelerate the diffusion of carbon during the CNM growth. We assume that the similar function could be also attributed to the tungsten. Based on results of our research, it is possible to suggest that, along with boosting impact on carbon diffusion, the addition of W leads to enhancement of catalytic function (decomposition of TCE molecules), which is evidences by the significant shortening of the induction period. In the recently reported paper we can find the confirmation of this assumption [6]. It was claimed that W is capable of increasing the catalytic activity and stability of nickel particles during the hydrocarbon decomposition due to the partial transfer of electrons from Ni to W, along with the formation of the W2C phase, which served as a regulator of carbon atoms from the surrounding atmosphere to Ni. The corresponding explanation has been added to the text of paper, including the conclusion section. [4] Potylitsyna, A. R., Bauman, Y. I., Mishakov, I. V., Plyusnin, P. E., Vedyagin, A. A., Shubin, Y. V.The Features of the CCVD of Trichloroethylene Over Microdispersed Ni and Ni–Mo Catalysts. Top. Catal. 2022, 1-12. [5] Modekwe, H. U., Mamo, M. A., Moothi, K., Daramola, M. O. Effect of different catalyst supports on the quality, yield and morphology of carbon nanotubes produced from waste polypropylene plastics. Catalysts. 2021, 11(6), 692. [6] Jia, J., Veksha, A., Lim, T. T., Lisak, G., Zhang, R., & Wei, Y. Modulating local environment of Ni with W for synthesis of carbon nanotubes and hydrogen from plastics. J. Clean. Prod. 2022, 352, 131620. As for the morphology and structure of carbon product, the comparison of numerous SEM and TEM images shows insignificant effect of W addition on it. We have selected more TEM data especially for your reference to support this conclusion (see Figure below). Figure 3. TEM images of the carbon product obtained by the decomposition of TCE at 600 °C over 100Ni and 96Ni-4W catalysts We consider this result to be very positive, since the unique segmented structure of CNF product remains preserved in case of W addition into Ni-alloy. At the same time, the presence of W makes Ni-catalyst more active, which allows one to produce higher amount of the desirable CNF material. Thank you once again for your careful consideration of our work and especially valuable comments that helped us to make paper better!

Reviewer 2 Report
Manuscript ID: materials-1985603
Type of manuscript: Article
Title: Features of Catalytic Decomposition of Trichloroethylene over Ni-W Catalyst to Produce Segmented Carbon Nanofibers
This manuscript discusses about the Catalytic Decomposition of Trichloroethylene over Ni-W Catalyst to produce Carbon Nanofibers
The manuscript has been written in a very nice and I strongly recommend it to be accepted for publications.
· Only one comment: author should add an equation to show the calculation for v (mg/min)
Author Response
The manuscript has been written in a very nice and I strongly recommend it to be accepted for publications. Only one comment:
Q1. Author should add an equation to show the calculation for v (mg/min)
A1. Agree. The necessary explanations have been made in the text of the article.
The reaction rate (v) was found from the slope of the kinetic curve. The slope coefficient was measured from the experimentally obtained graph of the product mass gain vs. time (see Fig. 1). For this purpose, a conditionally linear section of the kinetic curve was selected (within the range of 20-120 min), and the slope was calculated using the straightener function.
Figure 1. An example of calculation of the carbon deposition rate over Ni-catalyst (time interval: 20-120 min). Decomposition of TCE at T = 600 °C

Reviewer 3 Report
The modification effect of W on the catalytic activity of Ni during the decomposition of TCE into carbon nanomaterials was systematically studied. The catalytic pyrolysis of 96Ni-4W alloy and unmodified nickel (100Ni) in the temperature range of 475-725°C was studied systematically. The results show that tungsten greatly enhances the catalytic activity, and the lifetime of 96Ni-4W and 100Ni catalysts is compared in detail. The authors have a comprehensive characterization in the experiment; however, some problems should be improved. It can be acceptable for publication after revisions.
(1) In FIG. 2, the author only shows the magnification of the diffraction peak of 331 crystal plane. Do the diffraction peaks of other crystal planes have similar shifts? Does the offset of each diffraction peak change?
(2) The doping amount of W in 96Ni-4W sample is 4%, which needs to be further confirmed by ICP data.
(3) Some recently published references could be cited to enrich the introduction part, such as 10.1016/j.jechem.2021.08.066, 10.1002/aenm.202200855, 10.1002/anie.202206460.
(4) In Figure 7, graphs of the TEM with the larger magnification need to be shown for the reader to compare.
(5) In the figure of the article, a,b and c are numbered from left to right or from top to bottom. The authors should confirm the order as needed.
Author Response
The modification effect of W on the catalytic activity of Ni during the decomposition of TCE into carbon nanomaterials was systematically studied. The catalytic pyrolysis of 96Ni-4W alloy and unmodified nickel (100Ni) in the temperature range of 475-725°C was studied systematically. The results show that tungsten greatly enhances the catalytic activity, and the lifetime of 96Ni-4W and 100Ni catalysts is compared in detail. The authors have a comprehensive characterization in the experiment; however, some problems should be improved. It can be acceptable for publication after revisions.
Q1. In FIG. 2, the author only shows the magnification of the diffraction peak of 331 crystal plane. Do the diffraction peaks of other crystal planes have similar shifts? Does the offset of each diffraction peak change?
A1. Thank you for your comment! Yes, we have a shift for all the peaks in diffraction pattern of Ni–W alloy, in comparison with pure nickel. The reason for this phenomenon is the change in the value of the lattice parameter of the alloy in comparison with that of nickel. An increase in the lattice parameter of the alloy occurs due to the formation of a substitutional solid solution. But the extent of the shift for peaks in different angles is not similar. For cubic lattice, we have the following relationship between lattice parameter (a) and interplanar distance (d):
1/d2 = (h2 + k2 + l2)/a2,
h, k, l – Miller indexes.
And also, the relationship between the interplanar distance (d) and diffraction angle (q), which is known as Bragg law:
2d sinq = l,
l – a wavelength.
Thus, the differentiation of Bragg law equation gives:
Δa/a = Δd/d = -ΔΘCtgΘ
So we can see that with increasing of diffraction angle (θ) the value of difference between angles (Δθ) also increases.
Q2. The doping amount of W in 96Ni-4W sample is 4%, which needs to be further confirmed by ICP data.
A2. Agree. According to ICP-AES analysis data, the doping amount of W in the composition of 96Ni-4W was equal to 4.4 wt.%. The experimental section and results section of paper have been corrected accordingly.
Q3. Some recently published references could be cited to enrich the introduction part, such as 10.1016/j.jechem.2021.08.066, 10.1002/aenm.202200855, 10.1002/anie.202206460.
A3. Thank you for suggestion! All the proposed papers [9, 10 and 11] have been added to the introduction.
Q4. In Figure 7, graphs of the TEM with the larger magnification need to be shown for the reader to compare.
A4. Agree. The corresponding TEM images with the larger magnification have been added to the text of the manuscript. Thank you for your suggestion!
Q5. In the figure of the article, a,b and c are numbered from left to right or from top to bottom. The authors should confirm the order as needed.
A5. Agree. The numbering of the pictures has been changed to follow the uniform style (from left to right).

Round 2
Reviewer 3 Report
This manuscript has been largely improved, and can be published now.
Author Response
Dear Reviewer!
Thank you very much for your kind recommendation!